# The Influence of Hydrofluoric Acid Temperature and Application Technique on Ceramic Surface Texture and Shear Bond Strength of an Adhesive Cement

**DOI:** 10.3390/ma16124303

**Published:** 2023-06-10

**Authors:** Cristiana Cuzic, Anca Jivanescu, Radu Marcel Negru, Iosif Hulka, Mihai Rominu

**Affiliations:** 1Department of Prosthodontics, Faculty of Dental Medicine, University of Medicine and Pharmacy “Victor Babes”, 300041 Timisoara, Romania; pricop.cristiana@umft.ro; 2Research Center of Digital and Advanced Technique for Endodontic, Restorative, and Prosthetic Treatment (TADERP), 300041 Timisoara, Romania; 3Research Center in Dental Medicine Using Conventional and Alternative Technologies, School of Dental Medicine, University of Medicine and Pharmacy “Victor Babes”, 300070 Timisoara, Romania; radu.negru@upt.ro (R.M.N.); rominu.mihai@umft.ro (M.R.); 4Department of Mechanics and Strength of Materials, Politehnica University Timisoara, 300222 Timisoara, Romania; 5Research Institute for Renewable Energie, Politehnica University of Timisoara, G. Muzicescu 138, 300501 Timisoara, Romania; iosif.hulka@upt.ro; 6Department of Prosthesis Technology and Dental Materials, Faculty of Dentistry, University of Medicine and Pharmacy “Victor Babes”, 300041 Timisoara, Romania

**Keywords:** glass–ceramic, hydrofluoric acid etching, surface treatment, adhesion, scanning electron microscopy, surface roughness, shear bond strength

## Abstract

All-ceramic restorations are the foundation of modern esthetic dentistry. Clinical approaches for preparation, durability, aesthetics, and repair have been reformed by the idea of adhesive dentistry. The aim of the study and the objective question was to evaluate the impact of heated hydrofluoric acid pretreatment and the application technique’s influence on the surface morphology and roughness of leucite-reinforced glass–ceramic materials (IPS Empress CAD, Ivoclar Vivadent), which is fundamental for understanding the adhesive cementation process. Scanning electron microscopy was used to observe the effectiveness of the two HF (Yellow Porcelain Etch, Cerkamed) application techniques and the HF’s temperature impact on the surface topography of the ceramic. Based on surface conditioning methods, the adhesive cement (Panavia V5, Kuraray Noritake Dental Inc., Tokyo, Japan) was applied to the conditioned ceramic samples and light-cured. Shear bond strength values were correlated with the micro-retentive surface texture of the ceramic. With universal testing equipment at a 0.5 mm/min crosshead speed, SBS values between the resin cement and the ceramic material were assessed until failure. Analyzing the fractured surfaces of the specimens by digital microscopy, the failure modes were divided into three categories: adhesive, cohesive, and mixed failure. Analysis of variance (ANOVA) was used to statistically analyze the collected data. The results show that alternative treatment methods affected the material’s surface characteristics and have an influence on the shear bond strength.

## 1. Introduction

Recent advancements in digital dentistry have raised new challenges for dental professionals. Due to its benefits, including its efficiency, usability, and therapeutic quality, CAD/CAM technology is widely applied in daily dentistry practice [1]. This technology has several uses in the dental office and dental laboratory, including manufacturing indirect prosthodontic restorations such as inlays, veneers, crowns, fixed partial dentures, and implant abutments [2]. The desired strength and the enhanced esthetics of the prosthodontic restorations combined with the precise and convenient technological process define the development of CAD/CAM technology [3]. Today’s dentistry practice widely uses ceramic materials while creating fixed dental prostheses. CAD/CAM systems are the central processing method that has been developed because they provide higher process reliability, excellent cost-effectiveness, and a significant decrease in working time [4]. The basic categories for ceramic materials include those based on their composition (e.g., silica-based ceramics, oxide ceramics, resin-matrix ceramics) and those produced by layering, pressing, and CAD/CAM milling [5]. Ideally, clinicians want to use a material that has the appropriate aesthetics as well as good mechanical characteristics, can sustain high occlusal forces, and is as crack-propagation-resistant as possible [6]. Obviously, the mechanical characteristics of ceramics are directly influenced by their microstructure.

Ceramic restorations containing a glass phase must be acid etched before bonding in order to generate the optimized surface structure and increase resin cement adhesion [7]. The highest bond strength of an adhesive-resin cement to glass ceramics was achieved by treating the porcelain’s surface with 5% to 9.5% hydrofluoric acid, etching the tooth structure with 37% phosphoric acid, and using a silane coupling agent. The results of the acid treatment vary depending on the treated ceramic material, the conditioner concentration, and the etching duration [8]. Deep involuted areas allow the resin to flow and interlock, strengthening the bond to etched surfaces. Following the etching process, the restoration is immersed in water and cleaned in an ultrasonic cleaner for five minutes, air-dried, and then silane is applied to the intaglio surface [7]. Glass ceramics are fragile and have low flexural strength; therefore, definitive adhesive cementation using composite resin should be utilized to strengthen the restoration’s fracture resistance [9]. It has been demonstrated that adhesive cementation increases fracture stresses and lengthens the durability of the restoration [10]. It is recommended to utilize light-, dual-, and chemically polymerized composite resin materials with glass ceramics [11].

The retention and durability of indirect ceramic restorations depend heavily on the adhesive cement’s bond strength. Shear and micro tensile bond strength tests are most often performed to examine types of cement and their adhesives [12].

In order to assess the capacity to withstand stress produced by occlusal forces, the tests analyze the adhesion between the tooth or the ceramic and cement material. Shear bond strength values are influenced by the substrate material and surface morphology and vary from one test design to another; therefore, they cannot be classified as material properties [13].

Previous research studies have examined surface conditioning, the cementation protocol, or the potential failure of all-ceramic prosthodontic restorations at the level of various interfaces [14]. Despite the existing studies on adhesive cementation procedures utilized by dentists, new studies in the literature show improvements when adhesives are applied appropriately for all-ceramic restorations. The clinician should concentrate his attention on the tooth preparation design and the anticipated thickness of the restoration to minimize the clinical failure rate. Before applying the adhesive cement, the future bond interfaces need to undergo a thorough cleaning pretreatment, and surface conditioning in order to achieve high resistance to masticatory stresses and be as durable as possible over time [14]. Considering manufacturers’ attempts to create and distribute user-friendly self-etching adhesive types of cement, the question of whether dentists should choose these products over the traditional adhesive preparation with etching and priming still has to be answered [15].

This research aimed to investigate the effect of application method and temperature of preheated hydrofluoric acid pretreatment on the surface morphology of leucite-reinforced glass–ceramic materials (IPS Empress CAD, Ivoclar Vivadent), which is essential for comprehending the adhesive cementation process. Then, the values of the shear bond strength of an adhesive cement (Panavia V5, Kuraray Noritake Dental Inc., Tokyo, Japan) were associated with the ceramic’s retentive surface morphology.

## 2. Materials and Methods

### 2.1. Specimen Preparation and Surface Conditioning

Fifty blocks of leucite-reinforced glass–ceramic (IPS Empress, Ivoclar Vivadent, Schaan, Liechtenstein) specimens were polished using SiC papers with grit levels of #1200, #1500, and #2500 while the specimens were continuously cooled by water in a grinding machine (ECOMET Grinder/Polisher, Buehler, Uzwil, Switzerland) followed by 5 min immersion in an ultrasonic bath in distilled water. A 9.5% HF gel (Yellow Porcelain Etch, Cerkamed, Stalowa Wola, Poland) was preheated in an incubator (Ivoclar Vivadent Cultura) at 50 °C (for 20 min). The materials used in the present study are detailed in Table 1.

The ceramic blocks were randomly divided (n = 10) into five groups according to the surface treatments as follows:Group 1: NT (control group)—no surface treatment;Group 2: DH—dynamic application of preheated HF gel for 60 s of continuous movements of the micro brush on the ceramic surface;Group 3: SH—static application of preheated HF gel for 60 s without brushing;Group 4: DNH—dynamic application of nonheated HF gel (at room temperature) for 60 s of active movements with a micro brush on the surface;Group 5: SNH—static application with a micro brush of nonheated HF gel (at room temperature) for 60 s without brushing.

All specimens were rinsed with an air/water spray for 20 s immediately after the HF treatment and air dried for 10 s.

### 2.2. Scanning Electron Microscopy (SEM) of Surface Morphology

The etched surfaces of the specimens from each experimental group were analyzed for topographic patterns of the specimens using a scanning electron microscope (SEM Quanta FEG 250, FEI, Hillsboro, OR, USA) and a secondary electron detector (SE) to examine how the treatments affect the materials’ surface morphology. To avoid charging, the SEM was operated in low vacuum mode.

For visual inspection, SEM micrographs of each ceramic surface were collected at 1000× and 5000× magnifications using SEM to assess the morphological changes of the surface and to evaluate any impact on the ceramic area following the treatment of the specimens. Each specimen was analyzed in the center of the ceramic-conditioned surface.

### 2.3. Shear Bond Tests

According to the manufacturer’s instructions, a silane coupling agent (Clearfil Ceramic Primer Plus; Kuraray Noritake Dental Inc., Tokyo, Japan) was applied with an applicator brush on all specimens.

From a polyvinyl tube with an inner diameter of 3 mm and a height of 5 mm, translucent smaller cylinders were methodically cut with parallel ends. One cylinder was used for each specimen in order to bond the adhesive cement on the conditioned surfaces. Each polyvinyl cylindrical mold was gently filled with cement (Panavia V5, Kuraray Noritake Dental Inc., Tokyo, Japan) through the opening of the polyvinyl tube after being positioned across the surface of the treated specimen (Figure 1). As a result, cylinders of adhesive cement were bonded to the treated surfaces and light-cured using an LED curing device (DTE LUX-E Plus Curing Light, Woodpecker, power intensity 1000 mW/cm^2^) from two opposite sides for 20 s. Because of the tube’s thickness of 1 mm, the LED curing device was activated by being in contact with the tube. The polyvinyl tubes were removed after the adhesive was completely cured (using both light and chemicals). All of the specimens were stored in distilled water for seven days before the bond strength testing.

The Shear Bond Strength (SBS) tests were carried out using a Zwick/Roell ProLine Z005 universal testing machine (ZwickRoell, Ulm, Germany). The tests were performed at ambient temperature and a crosshead speed of 0.5 (mm/min).

The specimens were fixed using a precision vise (Figure 2), and the tester blade was placed at an angle of 90° to the adhesive cement cylinder. The specimens were subjected to shear loading along the interface until fracture, and the force was recorded through the TestXpert II software.

The shear strength τ_max_, expressed in MPa, was determined from the conventional formula:τ_max_ = F_max_/A
where F_max_ represents the maximum force recorded at failure, expressed in newton (N), and A is the shear area, i.e., the area of a circle having diameter d, expressed in (mm^2^).
A = π·d^2^/4

### 2.4. Statistical Analysis

The data were statistically analyzed using SPSS Statistics 29.0 software (IBM, New York, NY, USA, 2022) for a level of significance of α = 0.05.

SBS data were preliminarily tested for normality and homogeneity using the Shapiro–Wilk and Levene tests. The first null hypothesis, which claims that the variable SBS is normally distributed, could not be rejected. The second null hypothesis assumed that the variances of the five groups are equal, so they are not statistically significantly different.

Further, one-way analysis of variance (ANOVA) and the post hoc Tukey HSD test were applied to establish whether significant differences in terms of SBS occurred among the five groups. The null hypothesis that the mean SBS values for all five groups are equal was rejected.

A two-way ANOVA (general linear model) was performed for groups DH, SH, DNH, and SNH in order to examine the differential influence of the factors, temperature (heated or nonheated HF gel), and application regime (static or dynamic application) on the dependent variable SBS.

### 2.5. Digital Microscopy of Fracture Surfaces

A digital microscope in the university research center CMDTCA (Research Center in Dental Medicine Using Conventional and Alternative Technologies) was used to analyze the failure mechanisms at a 50× magnification for both interfaces, the ceramic and the adhesive cement. Failure types were divided into three categories: adhesive (A; failure at the bond interfaces where the ceramic and the resin cement substrate were connected), cohesive (C; failure of at least one of the substrates—the ceramic or the adhesive cement), and mixed (M; A + C).

## 3. Results

### 3.1. SEM Observations

The SEM example images with significant ceramic surface morphologies of all experimental groups are shown in Figure 3a–j.

Depending on the temperature of the HF and its application method, different patterns were seen. The most representative pattern for each specimen was selected to be analyzed. Without HF preparation, the specimen had a smooth surface texture lacking major ceramic structural imprints.

Compared to the porous surface of all the studied etched groups, the NT group displayed a less retentive pattern. It was found that all the HF treatments left the ceramic surface with significant porosities. Consequently, following surface treatments, the surface morphology of each CAD/CAM block significantly changed. On SEM micrographs, these changes in surface roughness were readily detectable.

The ceramic conditioning method had an influence on the ceramic surface micro retentions, according to the results from the microscopic study.

### 3.2. SBS Test Results

The descriptive statistics for SBS, expressed in (MPa), including the means and standard deviations for the five groups, are listed in Table 2.

Figure 4 presents the SBS box plot results. Differences in SBS were observed between the groups. The SNH and DNH groups showed the highest SBS values compared to the NT and DH groups, for which the SBS values were the lowest. The SH group presented an intermediate mean value for SBS.

### 3.3. Statistical Analysis

The Shapiro–Wilk test results, statistics, and significance are presented in Table 3. The null hypothesis, which states that the variable SBS is normally distributed, could not be rejected (*p* > 0.05) for any of the groups. Thus, the variable SBS may be normally distributed for all five groups.

For the Levene test, the significance of 0.148 was greater than the defined level of 0.05, and thus the null hypothesis was maintained and there was no difference between the variances of the five groups.

Following the tests of normality and homogeneity, one-way ANOVA [16] established that there are significant differences among the groups in terms of SBS (Table 4), i.e., it can be inferred that within the groups, the mean values are not equal. The test statistic F was greater than the critical value Fcrit, and its significance was *p* < 0.05. The null hypothesis that the mean SBS values for all five groups are equal, was rejected.

The post hoc Tukey HSD test revealed that differences in terms of SBS occurred among the five groups, as is listed in Table 5 [17].

There were statistically no differences between groups NT, DH, and SH; between groups SH and DNH; or between groups DNH and SNH. The differences were statistically significant between group NT and group DNH; group NT and group SNH; group DH and group DNH; group DH and SNH; group SH and SNH.

The results of the two-way ANOVA are presented in Table 6. It can be seen that both parameters, temperature and application regime, had a statistically significant influence on the SBS. Looking at the partial Eta squared values, it can be concluded that the effect of temperature was higher than the effect of the application regime. Instead, the interaction effect was not significant for the SBS dependent variable.

### 3.4. Digital Microscopy Examination

After performing the SBS tests, the ceramic surfaces were examined with a digital microscope in order to assess the types of fractures at the interface.

Adhesive failures occurred at the bond interfaces (Figure 5) between the ceramic and adhesive cement substrates. Three types of failure were identified: adhesive (failure at the bonding interface), cohesive (fracture within ceramic material), and mixed. The mixed type was shown to be the predominant failure mode.

## 4. Discussion

The shear bond strength values of the ceramic material to resin cement were improved by static surface treatment techniques. The two influencing factors—temperature of the hydrofluoric acid (heated or not heated) and application mode (static or dynamic)—had statistically significant effects on the SBS. In contrast to the technique of application, the applied HF temperature had a significant impact.

The interaction of these two factors did not significantly influence the results in terms of SBS shear strength.

There was no statistically significant difference in the mean SBS values for the NT, DH, and SH preparations. In contrast to the NT control group, the surface preparations of the DNH and SNH groups improved the average SBS values by 56.32% and 74.88%, respectively. The influence of the two factors can also be observed in the increase in SBS values for DNH and SNH conditions compared to the DH group, at 65% and 84.59%, respectively. The SH and DNH groups’ preparations did not present statistically distinctive SBS shear strength results due to the opposing effects of the two influencing factors; however, the SNH group compared to SH indicated a 35.73% improvement in SBS. The shear strengths of the DNH and SNH groups, which varied exclusively in how the HF was applied, were not noticeably different.

The adhesion between the ceramic and resin cement is crucial for the long-term performance of all ceramic-based dental restorations. Due to poor adhesion, the restoration may fail as a result of the fracture that developed in the restorative material or along the cement interface [18]. Roughening the intaglio surfaces of all ceramic restorations through etching has reportedly increased adhesion surface area, making the bond between the ceramic surface and resin-based materials possible.

The acid preferentially etches the crystalline or amorphous phases of the ceramic, producing unsaturated oxygen bonds [19] that act as bonding partners for phosphate monomers with dual functions [20,21]. Hydrofluoric acid generates porous uneven surfaces and micro retention sites by selectively dissolving the glassy or crystalline matrix of the ceramic material. These microporous ceramic surfaces expand their surface area and make it easier for the resin to penetrate them. According to Sorenson et al. [22], using HF etching on feldspathic ceramics significantly increases the bond strength.

In order to optimize micromechanical retention and the wettability of the applied primer, which leads to higher bond strength values, chemical conditioning (acid etching) increases the surface roughness and surface energy of the ceramic materials. The interfacial tension between the material and the adhesive, as well as the material’s surface energy, have been discovered to be the key determinants of bond strength values [23,24].

Although surface roughness and surface energy are not directly correlated, higher surface energy results in higher bond strength values [25]. Şişmanoğlu et al., found that the HF treatment and silanization combination resulted in the highest bond strength values for feldspathic ceramic [26,27]. The most favorable surface treatment for leucite-based ceramics is etching with hydrofluoric acid associated with silan [28].

Hydrofluoric acid has started to be utilized for conditioning the surfaces of restorative materials since the development of glass-based ceramics and recognition of the benefits of adhesive cementation in dentistry [29]. Universal adhesives improve clinical application processes and make them easily for practitioners to use [30].

The outcomes noted in similar studies have pointed out that mechanical interlocking with the roughness produced on the ceramic surface is the primary determinant for a strong bond, even though a different silane treatment is necessary, especially for feldspathic ceramics [31,32]. Roughness is a vital surface property of restoring materials, influencing the substances’ abrasiveness and mechanical retention despite the stresses from the external environment. Surface roughness is not the only determinant of material adhesion; it is also influenced by other characteristics, such as porosity, residual microstructural tension, composition, and mass defects [33].

There are various in vitro techniques that can be used to assess the bond strength between two substrates (shear, micro shear, and tensile). The basic idea behind these tests is to load the specimen with forces that cause stress at the adhesive interface until specimen failure is seen. These tests have benefits and drawbacks, but none is acknowledged as a universal approach. Moreover, factors like cross-head speed, sample shape, and substrate brittleness may have an impact on the results [34,35]. The critical load measured for the shear bond tests was unable to accurately reflect the bond strengths attained by the various surface treatments at the adhesive interface.

As compared to chemical surface treatments performed by other ceramic primers, Queiroz et al., reported stronger shear bond strengths on ceramic surfaces treated with hydrofluoric acid. They determined that using primers alone was insufficient to create an adequate bond strength to feldspathic ceramic and that acid etching, combined with ceramic primer, was required to achieve an adequate bond strength between glass ceramic and composite resin [36].

Nonetheless, Şişmanoğlu et al. [37] highlighted that the combination of surface conditioning and further silane application produced the best bond strength values. When the quantity of ceramic in the restorative material’s composition increases, the HF treatment produces stronger bonds, whereas the amount of polymer in hybrid materials generates stronger bonds when exposed to airborne particle abrasion. When silane is applied to glass ceramic, both HF etching and airborne particle abrasion can be performed. When the silane-containing adhesive is used by itself, HF etching is indicated.

CAD/CAM technology advancement has resulted in shorter turnaround times, fewer expenses, and better patient results. Furthermore, digital dentistry technology allows clinicians to provide same-day restorations to their patients, eliminating the need for several sessions and interim restorations. As a consequence, new ceramic materials have been designed expressly for use with CAD/CAM systems. These biomaterials are created to satisfy the specific needs of dental restorations, such as strength, durability, and biocompatibility, and to optimize even more complex treatment plans like full mouth rehabilitations or implant-prosthetic restorations [38]. Overall, using CAD/CAM technology in dentistry has significantly enhanced the quality of care that dentists can deliver to their patients while simplifying the workflow in laboratories and clinics. The patients’ personal aesthetic evaluations in Capparé et al. [39] revealed good outcomes that were consistent with those published in the literature. In particular, the patient’s contentment assessment revealed that the digital protocol provided more satisfaction in terms of comfort/discomfort.

It should be mentioned that the SBS test’s limitations in reproducing clinical loading forces and the simulation of aging within the oral environment may affect this in vitro study. Further research is recommended to analyze the ceramic surface treatment and its clinical impact on the bond strengths within the context of the most favorable ceramic surface conditioning.

The dentist’s ability to choose the appropriate restoration material, manufacturing process, and cementation or bonding techniques according to intraoral conditions and aesthetics will be essential to the success of the prosthodontic treatment. The surface treatment conditioning and the continuous refinement of the cementation protocols of dental restorations will guide clinical performance.

## 5. Conclusions

Within the limitations of this study, the following conclusion can be drawn:Both factors (HF temperature and its application technique) significantly affect SBS values; moreover, the temperature is a more influencing parameter. Compared to the control group, the other four types of ceramic treatments improved the shear bond strength values.The hydrofluoric acid temperature and application technique determine the different ceramic surface patterns.The examination of interfaces after debonding revealed three different types of bonding failures: adhesive, cohesive, and mixed; the cohesive type of failure occurred exclusively in the ceramic material, and there was no cohesive fracture in the adhesive cement.

## Figures and Tables

**Figure 1 materials-16-04303-f001:**
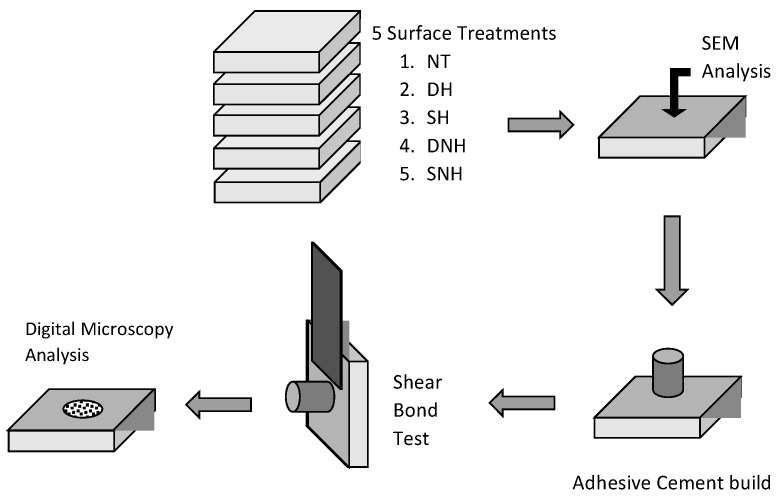
The study’s conceptual design.

**Figure 2 materials-16-04303-f002:**
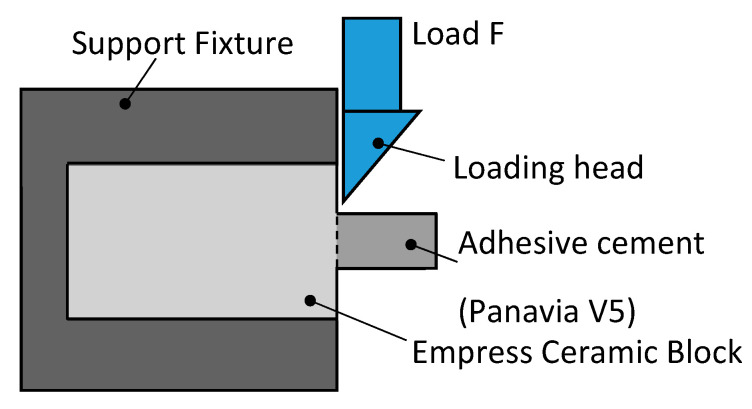
Illustration of the SBS test.

**Figure 3 materials-16-04303-f003:**
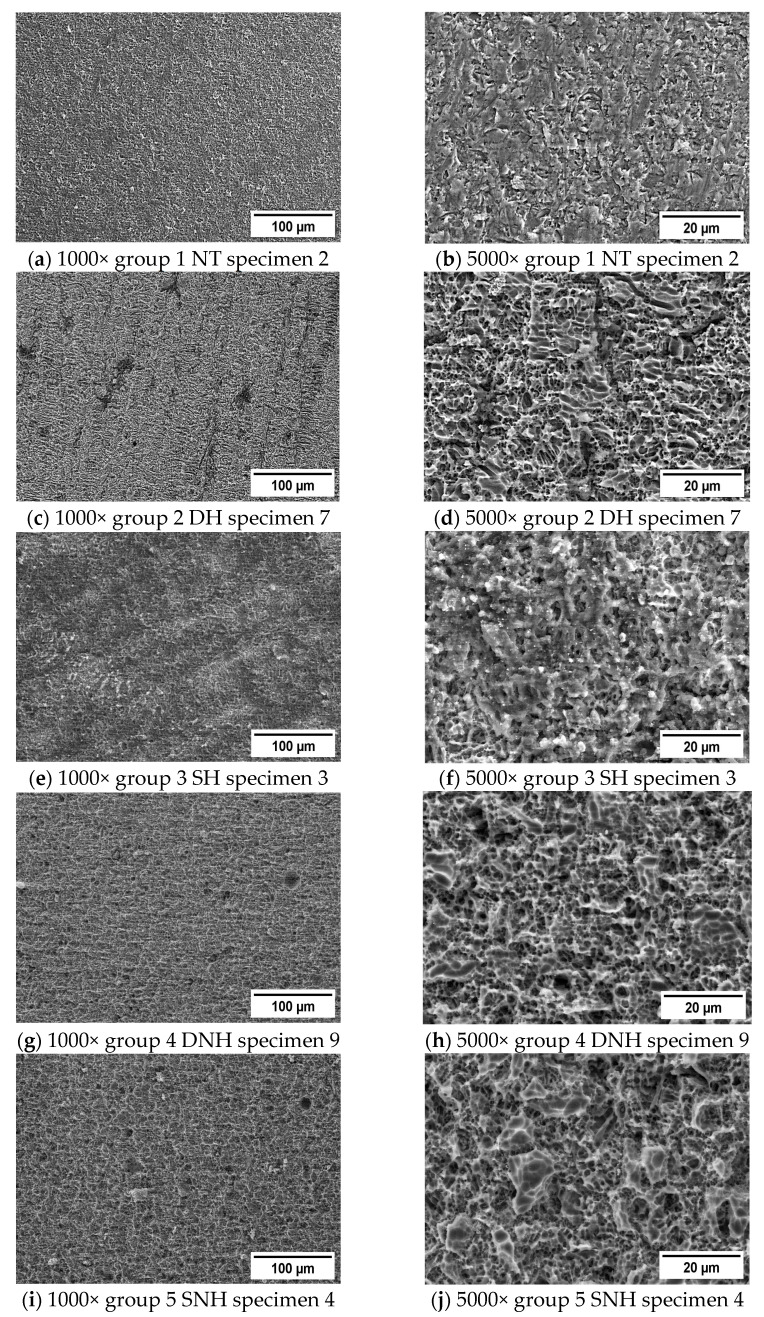
SEM micrographs of IPS Empress CAD after (**a**,**b**) no surface treatment/control group (NT); (**c**,**d**) dynamic application of heated HF (DH); (**e**,**f**) static application of heated HF (SH); (**g**,**h**) dynamic application of nonheated HF (DNH); and (**i**,**j**) static application of nonheated HF (SNH).

**Figure 4 materials-16-04303-f004:**
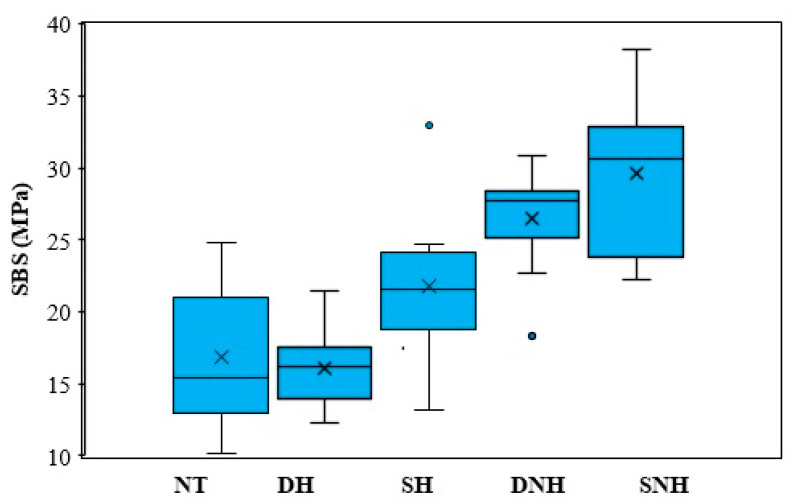
Box plot for the five groups.

**Figure 5 materials-16-04303-f005:**
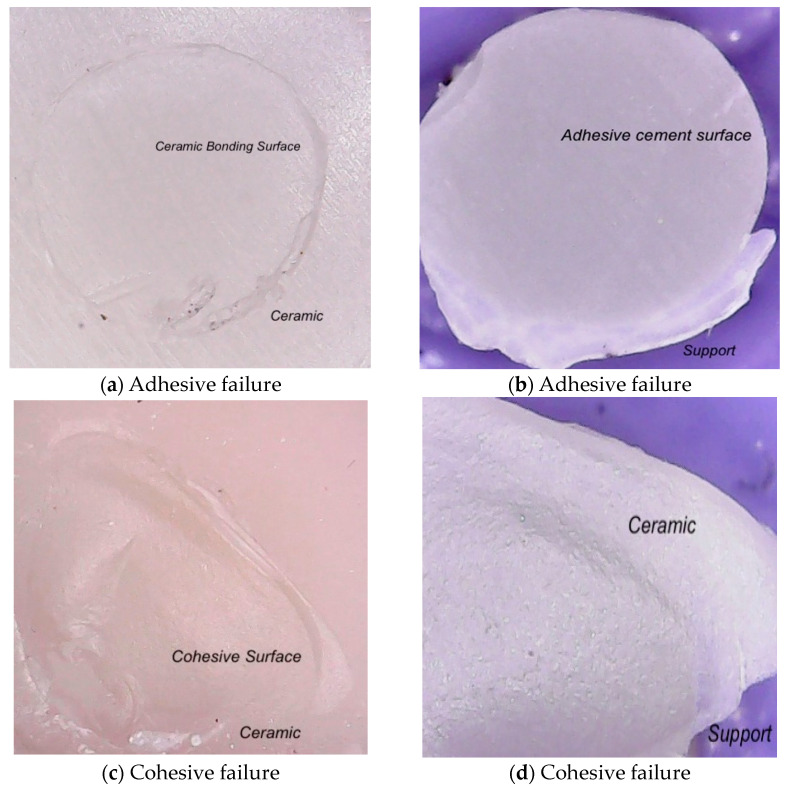
Images showing the typical fractographic characteristics after the shear test: (**a**) adhesive failure on the ceramic surface; (**b**) adhesive failure on the cement surface; (**c**) cohesive failure in the ceramic material; (**d**) cohesive failure aspect of the fractured ceramic; (**e**) mixed failure in the ceramic; (**f**) mixed failure aspect with a fracture in the ceramic layer.

**Table 1 materials-16-04303-t001:** Materials used in the study.

Material	Type	Manufacturer	Lot No.
Empress CAD	Leucite-reinforced glass ceramic CAD/CAM block	Ivoclar-Vivadent Schaan Liechtenstein	V01607, U50932, S49744, X11916
Yellow Porcelain Etch	Ceramic Etching gel	PPH Cerkamed Wojciech Pawlowski Stalowa Wola, Poland	2106181
Clearfil Ceramic Primer Plus	Ceramic primer	Kuraray Noritake Dental, Tokyo, Japan	760065
Panavia V5	Dual-cure resin cement	Kuraray Noritake Dental, Tokyo, Japan	4E0086

**Table 2 materials-16-04303-t002:** Descriptive statistics for the tested five groups.

Groups	N ^1^	Mean	SD ^2^	Min ^3^	Median	Max ^4^
NT	10	16.92	5.18	10.21	15.44	24.85
DH	10	16.03	2.86	12.23	16.25	21.51
SH	10	21.80	5.28	13.17	21.62	33.01
DNH	10	26.45	3.68	18.30	27.71	30.82
SNH	10	29.59	5.81	22.25	30.65	38.18

^1^ group size; ^2^ standard deviation; ^3^ minimum value; ^4^ maximum value.

**Table 3 materials-16-04303-t003:** Shapiro–Wilk and Levene test results.

Groups	Shapiro–Wilk Test		Levene Test			
	W ^1^	*p*	F ^2^	df1	df2	*p*
NT	0.928	0.428	1.787	4	45	0.148
DH	0.957	0.753				
SH	0.947	0.639				
DNH	0.898	0.210				
SNH	0.906	0.255				

^1^ Shapiro–Wilk test statistic; ^2^ Levene test statistic.

**Table 4 materials-16-04303-t004:** One-way ANOVA results.

Experimental Groups	SS ^1^	df ^2^	MS ^3^	F ^4^	*p*	F_crit_
Between groups	1386.440	4	346.610	15.711	<0.001	2.579
Within groups	992.745	45	22.061			
Total	2379.185	49				

^1^ Sum of squares; ^2^ degrees of freedom; ^3^ mean square; ^4^ test statistic.

**Table 5 materials-16-04303-t005:** The post hoc Tukey HSD results.

Groups		Mean Difference	*p*
NT	DH	0.883	0.993
	SH	−4.887	0.155
	DNH	−9.529 ^a^	<0.001
	SNH	−12.688 ^a^	<0.001
DH	SH	−5.770	0.063
	DNH	−10.412 ^a^	<0.001
	SNH	−13.551 ^a^	<0.001
SH	DNH	−4.642	0.195
	SNH	−7.781 ^a^	0.005
DNH	SNH	−3.139	0.571

^a^ statistically significant difference *p* < 0.05.

**Table 6 materials-16-04303-t006:** Two-way ANOVA results.

Source of Variation	SS ^1^	df ^2^	MS ^3^	F ^4^	*p*	η ^5^
** *Main effects* **						
Temperature	827.463	1	827.463	39.657	<0.001 ^a^	0.524
Application regime	198.426	1	198.426	9.510	0.004 ^a^	0.209
** *Interaction effects* **						
Temperature × Application regime	17.305	1	17.305	0.829	0.369	0.023
** *Error* **	751.159	36	20.866			

^1^ Type III sum of squares; ^2^ degree of freedom; ^3^ mean square; ^4^ test statistic, F_crit_ is equal to 4.113; ^5^ partial Eta squared; ^a^ statistically significant effect (*p* < 0.05).

## Data Availability

All data are available upon request.

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
