# Peer review of "The Influence of Hydrofluoric Acid Temperature and Application Technique on Ceramic Surface Texture and Shear Bond Strength of an Adhesive Cement"

_materials, 2023, doi:10.3390/ma16124303_

Round 1

Reviewer 1 Report

The study is genuine and very interesting, however the authors should address the following issues to improve the quality of the manuscript:

- Please add a short statement in the abstract to explain the outstanding research question.

- The introduction section is good, however the authors should include the null hypothesis/hypotheses at the end of it.

- Materials and machines should be listed in a table with complete information including reference/lot/model numbers.

- Line 136: was there a reference point were to analyze the specimen under SEM?

- Line 147: what do authors mean by this statement? "after being adjusted across the surface of the treated specimen". Adjusted in which aspect?

- Line 149 and 150: Please mention the curing distance.

- Thermal cycling was not done prior to mechanical testing, why?

- Figure 1 is excellent.

- All used softwares should be listed with their complete info and version numbers.

- Figure 2 is fuzzy, please improve its resolution.

- Figure 3 is excellent and supports the scientific facts.

- Figure 4: please explain the high levels of standard deviations noted in the bar graph (groups 1, 3, and 5).

- Figure 5: the quality of the images are not good because of the background. Please replace those images with high resolution ones.

- Please add the study limitations and directions for future research in the discussion section.

- The conclusion section was done nicely. 

Author Response

Dear Reviewer,

Thank you for your report and your indications.

- Please add a short statement in the abstract to explain the outstanding research question.

Lines 22-23 in the abstract were reformulated and the aims were stated to include the research question.

- The introduction section is good, however the authors should include the null hypothesis/hypotheses at the end of it.

Considering your suggestion, we were urged to modify Lines 178, 183, 237, and 250 to describe and highlight the null hypotheses.   

- Materials and machines should be listed in a table with complete information including reference/lot/model numbers.

The materials used in the present study are detailed in Table 1 (Line 112). We regretfully noticed that in the first submitted version of our manuscript, we mentioned another material from our research center used in a previous study. We cautiously did all the modifications including the HF Yellow Porcelain Etch, Cerkamed.

- Line 136: was there a reference point were to analyze the specimen under SEM?

Each ceramic surface had the same size (12.0 x 14.0 mm) and was analyzed in the center of the area conditioned with HF.

The area of interest was evaluated at the intersection of the diagonals of the surface.

- Line 147: what do authors mean by this statement? "after being adjusted across the surface of the treated specimen". Adjusted in which aspect?

We modified the context accordingly - “After being positioned on the surface of the treated specimen, each polyvinyl cylindrical mold was gently filled with adhesive cement” (Line 147).

- Line 149 and 150: Please mention the curing distance.

Because of the polyvinyl tube's thickness of 1 mm, the LED curing device was activated by being in contact with the tube, from two opposite sides for 20 s. (Line 152)

- Thermal cycling was not done prior to mechanical testing, why?

The samples from this research were not thermocycled since this was an initial study in which thermocycling was not incorporated. This process will be included in future studies.

All of the specimens were stored in distilled water for seven days before the bond strength testing (Line 154-155).

- All used softwares should be listed with their complete info and version numbers.

We completed Lines 175-176 accordingly. (SPSS Statistics 29.0 software IBM, New York, USA, 2022)

- Figure 2 is fuzzy, please improve its resolution.

We modified the resolution and size of Figure 2.

- Figure 4: please explain the high levels of standard deviations noted in the bar graph (groups 1, 3, and 5).

Even if the standard deviations appear to be different in absolute value, the homogeneity test Levene did not reject the null hypothesis, meaning the five groups' variances are statistically equal. (Table 3.)

The levels of standard deviations are a specific scattering of any experimental data of research trials, in this situation without statistically significant differences.

- Figure 5: the quality of the images are not good because of the background. Please replace those images with high resolution ones.

The purple background represents the color of the support on which the specimens were mounted for analysis.

We managed to reduce the area of the visible background so that the light of the background isn’t reflecting and affecting the quality of the figures. (Figure 5.)

- Please add the study limitations and directions for future research in the discussion section.

The study limitations and future directions can be found in the discussion section (Lines 376-380).

We look forward to hearing from you in due time regarding our submission and responding to any further questions and comments you may have.

Sincerely,

Dr. Cuzic

Reviewer 2 Report

Dear Authors,

the paper is very interesting and can be considered for publication after minor revisions. In particular some aspects should be improved:

1) Please discuss the potential role of such system when using digital workflow, also in implantology. Please cite DOI10.3390/biology10121281

2) Please discuss if such system can improve or change the prognosis of implant-prosthetic rehabilitation on implants with different connections. Please cite DOI10.3390/biomedicines110411283) Please discuss if such system can improve the prognosis of full-arch rehabilitations. Discuss and cite PubMed ID29682552

English is good

Author Response

Dear Reviewer,

Thank you for your report and your indications.

1) Please discuss the potential role of such system when using digital workflow, also in implantology. Please cite DOI10.3390/biology10121281

2) Please discuss if such system can improve or change the prognosis of implant-prosthetic rehabilitation on implants with different connections. Please cite DOI10.3390/biomedicines110411283) Please discuss if such system can improve the prognosis of full-arch rehabilitations. Discuss and cite PubMed ID29682552

We agree with this and have included your suggestions throughout the revised manuscript Lines 364-377. The mentioned citations are also listed on page 14.

We look forward to hearing from you in due time regarding our submission and responding to any further questions and comments you may have.

Sincerely,

Dr. Cuzic

Reviewer 3 Report

IN LINE 62 IT TALKS ABOUT FELDESPATHIC CERAMICS BUT ACCORDING TO THE PROPORTION OF ACID I AM NOT SURE IF IT IS CORRECT SINCE IF IT IS FELDESPATHIC THE PERCENTAGE IS 9% BUT IF THE CERAMIC IS DISILIQUE IT WOULD BE 5%. PLEASE OR ADD THE DISILICATE CERAMIC OR REMOVE THE PERCENTAGE OF 5%.

I DON'T UNDERSTAND WHY THEY DON'T USE A NATURAL SUBSTRATE (A NATURAL TOOTH REMOVED) TO MEASURE ADHESION SINCE, UNLESS THEY JUSTIFY ME, A POLYVINYL TUBE WILL NOT BEHAVE THE SAME AS A NATURAL SUBSTRATE AND THEREFORE THESE RESULTS CANNOT BE EXTRAPOLATED TO REALITY.

Author Response

Dear Reviewer,

Thank you for your report and for pointing this out.

1st Comment: IN LINE 62 IT TALKS ABOUT FELDESPATHIC CERAMICS BUT ACCORDING TO THE PROPORTION OF ACID I AM NOT SURE IF IT IS CORRECT SINCE IF IT IS FELDESPATHIC THE PERCENTAGE IS 9% BUT IF THE CERAMIC IS DISILIQUE IT WOULD BE 5%. PLEASE OR ADD THE DISILICATE CERAMIC OR REMOVE THE PERCENTAGE OF 5%.

Response: We agree with this comment. Therefore, we revised and changed the content of lines 60-63 referring to the category of glass ceramics which are etched with hydrofluoric acid between 5%-9,5% depending on the specific ceramic composition.

2nd Comment: I DON'T UNDERSTAND WHY THEY DON'T USE A NATURAL SUBSTRATE (A NATURAL TOOTH REMOVED) TO MEASURE ADHESION SINCE, UNLESS THEY JUSTIFY ME, A POLYVINYL TUBE WILL NOT BEHAVE THE SAME AS A NATURAL SUBSTRATE AND THEREFORE THESE RESULTS CANNOT BE EXTRAPOLATED TO REALITY.

Response: Thank you for this suggestion. It would have been interesting to explore this aspect as well. However, in the case of our study, we focused on the adhesion between ceramic and resin cement.

We have, accordingly, modified Line 153 in the revised version of our manuscript to emphasize this point by mentioning that the polyvinyl tubes were removed after the complete cure of the adhesive material. The polyvinyl tubes were used as a mold for the adhesive cement and were not involved in the evaluation of bond strength.

Further research will be required to examine the ceramic surface treatment and its clinical influence on bond strengths in the context of the best ceramic surface conditioning, considering the interface to a natural tooth substrate (Lines 378-383).

We look forward to hearing from you in due time regarding our submission and responding to any further questions and comments you may have.

Sincerely,

Dr. Cuzic

Round 2

Reviewer 3 Report

AFTER THE CORRECTIONS MADE AND HAVING EXPLAINED AND EMPHASIZED IN THE TEXT THAT WHAT IS MEASURED IS THE UNION OF THE CEMENT WITH THE CERAMIC. I GIVE MY APPROVAL TO THE PUBLICATION OF THE ARTICLE BUT I RECOMMEND FOR FUTURE RESEARCH THE USE OF NATURAL TEETH SAMPLES.